# SELF-INDUCED CURRICULUM LEARNING IN NEURAL MACHINE TRANSLATION

## ABSTRACT

Self-supervised neural machine translation (SS-NMT) learns how to extract/select suitable training data from comparable —rather than parallel— corpora and how to translate, in a way that the two tasks support each other in a virtuous circle. SS-NMT has been shown to be competitive with state-of-the-art unsupervised NMT. In this study we provide an in-depth analysis of the sampling choices the SS-NMT model takes during training. We show that, without it having been told to do so, the model selects samples of increasing (*i*) complexity and (*ii*) task-relevance in combination with (*iii*) a denoising curriculum. We observe that the dynamics of the mutual-supervision of both system internal representation types is vital for the extraction and hence translation performance. We show that in terms of the human Gunning-Fog Readability index (GF), SS-NMT starts by extracting and learning from Wikipedia data suitable for high school (GF=10–11) and quickly moves towards content suitable for first year undergraduate students (GF=13).

## 1 INTRODUCTION

Human learners, when faced with a new task, generally focus on simple examples before applying their gained knowledge on more complex instances. This approach to learning based on sampling from a curriculum of increasing complexity has also been shown to be beneficial for machines and has been named *curriculum learning* (CL) (Bengio et al., 2009) by the machine learning community. Previous research on curriculum learning has focused on selecting the best distribution of data — i.e. order, difficulty and closeness to the final task— to train a system. In this approach, data is prepared for the system to ease the learning task. In this work, we follow a complementary approach: we design a system that selects by itself the data to be trained on, and we analyse the selected distribution of data —order, difficulty and closeness to the final task— without imposing it beforehand. Our method resembles *self-paced learning* (SPL) (Kumar et al., 2010), in that it uses the emerging model hypothesis to select samples online that fit into its space as opposed to most curriculum learning approaches that rely on judgements by the target hypothesis, i.e. an external *teacher* (Hacohen & Weinshall, 2019) to design the curriculum.

The task explored in our work is machine translation (MT). In particular, we focus on self-supervised machine translation (SS-NMT) (Ruiter et al., 2019), which exploits the internal representations of an emergent neural machine translation (NMT) system to select useful data for training, where each selection decision is dependent on the current state of the model. Self-supervised learning (Raina et al., 2007; Bengio et al., 2013) involves a primary task (PT), for which labelled data is not available, and an auxiliary task (AT) that enables the PT to be learned by exploiting supervisory signals within the data. In the case of SS-NMT, both tasks —data extraction and learning NMT— enable and enhance each other, such that this mutual supervision leads to a self-induced curriculum, which is the subject to our analysis.

Curriculum learning has been widely studied. In Section 2 we describe the related work on CL, especially focusing on MT. Section 3 introduces the main aspects of self-supervised neural machine translation. Here, we analyse the quality of both the primary and the auxiliary tasks. This is followed by a detailed study of the self-induced curriculum in Section 4 where we analyse the characteristics of the distribution of training data obtained in the auxiliary task of the system. Finally, we draw our conclusion in Section 5.

## 2 RELATED WORK

In recent years, **machine translation** (MT) has experienced major improvements in translation quality by the introduction of neural architectures including RNNs (Cho et al., 2014; Bahdanau et al., 2014) and transformers (Vaswani et al., 2017). However, these rely on the availability of large amounts of parallel data. To overcome the need for labelled data, unsupervised neural machine translation (Lample et al., 2018a; Artetxe et al., 2018b; Yang et al., 2018) focuses on the exploitation of vast amounts of monolingual data by combining denoising autoencoders with back-translation and multilingual encoders. Further combining these with phrase tables from statistical machine translation (SMT) leads to impressive results (Lample et al., 2018b; Artetxe et al., 2018a; Ren et al., 2019; Artetxe et al., 2019). Nevertheless, as these still rely on hundreds of millions of monolingual sentences —which are not easy to come by for most languages— alternative methods focusing on the exploitation of smaller amounts of comparable corpora have recently been introduced in the form of self-supervised NMT (Ruiter et al., 2019). Here, the internal representations of an emergent NMT system are used to identify useful sentences in aligned documents. As the selection is dependent on the current state of the model, it resembles a type of self-paced learning Kumar et al. (2010). Also Wu et al. (2019) exploit comparable corpora and include similar sentences in NMT training. A comparison between both approaches is given in Section 3.1.

The data selection in SS-NMT is directly related to **curriculum learning**, which, in its essence, is the idea of presenting training samples in a *meaningful* order to benefit the learning, e.g. in the form of faster convergence or better performance (Bengio et al., 2009). Inspired by human learners, Elman (1993) argue that a neural network's optimization can be accelerated by providing samples in order of increasing complexity. While sample difficulty is an intuitive measure on which to base a learning schedule on, a variety of curricula focus on other metrics such as task-relevance or noise.

Up to now, **curriculum learning in NMT** has had a strong focus on the relevance of training samples to a given translation task, e.g. domain adaptation, where the task is the optimization for a specific domain. van der Wees et al. (2017) train on increasingly relevant samples by the gradual exclusion of irrelevant ones. In line with the basic hypotheses of CL, they observed an increase in BLEU over a static NMT baseline and a significant speed-up in training as the data size is incrementally reduced. Reversely, Zhang et al. (2019) adapt an NMT model to a domain by introducing increasingly domain-distant (*difficult*) samples. This contradictory behavior of benefiting from both increasingly difficult (domain-distant) *and* easy (domain-relevant) samples has been analyzed by Weinshall et al. (2018), showing that the initial phases of training benefit from easy samples w.r.t. the target hypothesis, while also being *boosted* (Freund & Schapire, 1996) by samples that are difficult w.r.t. the current hypothesis (Hacohen & Weinshall, 2019). Such boosting methods, where the model focuses longer on difficult samples, have been applied to NMT by Zhang et al. (2017). In a more complex setup, Wang et al. (2019b) adapt an NMT model to several domains by introducing samples with increasing relevance to all selected domains. In Wang et al. (2019a) both domain-relevance and NMT denoising are combined into a single curriculum.

The denoising curriculum for NMT proposed by Wang et al. (2018) is quite related to our approach in that they also apply an *online data selection* approach to build the curriculum based on the current hypothesis of the model. However, the noise scores for the dataset at each training step depend on fine-tuning the model on a small selection of clean data, which comes with a high computational cost. To alleviate this cost, Kumar et al. (2019) use reinforcement learning on the pre-scored noisy corpus to jointly learn the denoising curriculum with NMT. In Section 3.2 we show that our model exploits its self-supervised nature to perform denoising by selecting parallel pairs with increasing accuracy —without the need of additional noise metrics.

Difficulty-based curricula for NMT that take into account sentence length and vocabulary frequency have been shown to improve translation quality when samples are presented in increasing complexity Kocmi & Bojar (2017). Platanios et al. (2019) link the introduction of difficult samples with the NMT models' *competence*. Other difficulty-orderings have been explored extensively in Zhang et al. (2018), showing that they, too, can speed-up training without a loss in translation performance.

As a by-product, SS-NMT extracts cross-lingual similar sentence pairs, so the extractions can be compared to **parallel data mining** systems where strictly parallel sentences are expected. Early approaches exploited structural information of web-pages (Resnik & Smith, 2003) or Wikipedia (Adafre & de Rijke, 2006). Others relied on statistical measures such as cross-lingual information retrieval (Utiyama & Isahara, 2003), maximum entropy classifiers (Munteanu & Marcu, 2005), conditional random fields (Smith et al., 2010), SMT or a combination of methods (Fung & Cheung,

| | L1 (*en*) | | | L2 (*fr/de/es*) | | |
|---|---|---|---|---|---|---|
| | # Sents (M) | # Tokens (M) | ∅ | # Sents (M) | # Tokens (M) | ∅ |
| WP$_{enfr}$ | 117/42 | 2693/1205 | 28 | 38/25 | 644/710 | 16 |
| WP$_{ende}$ | 117/37 | 2693/987 | 29 | 51/30 | 1081/742 | 24 |
| WP$_{enes}$ | 117/35 | 2693/937 | 32 | 27/20 | 691/572 | 17 |
| EP$_{enfr}$ | 1+6 | 25+80 | 28 | 1+3 | 27+87 | 16 |
| EP$_{ende}$ | 1+9 | 25+180 | 29 | 1+7 | 26+192 | 24 |
| EP$_{enes}$ | 1+7 | 24+84 | 32 | 1+4 | 26+91 | 17 |

Table 1: Number of sentences and tokens for all corpora used. For WP, we report the sizes for both the monolingual/comparable editions. Sizes for pseudo-comparable EP corpora are reported for both true+false splits. Average sentences per article (∅) is given for all comparable corpora.

2004; Yasuda & Sumita, 2008; Abdul-Rauf & Schwenk, 2009; Barrón-Cedeño et al., 2015). Nowadays, sentence representations obtained from NMT systems or devoted architectures are achieving a new state of the art on parallel sentence extraction and filtering, see for instance (España-Bonet et al., 2017; Grégoire & Langlais, 2018; Schwenk, 2018; Bouamor & Sajjad, 2018; Artetxe & Schwenk, 2019a; Hangya & Fraser, 2019; Chaudhary et al., 2019). Due to its multilingual aspect, we consider the state-of-the-art method used in Schwenk et al. (2019) as a comparison point (see Section 3.2).

## 3  SELF-SUPERVISED NEURAL MACHINE TRANSLATION, SS-NMT

SS-NMT is a joint data selection and training framework for machine translation, which was originally introduced in Ruiter et al. (2019). It enables the learning of NMT on *comparable* rather than parallel data; where comparable data is a collection of multilingual topic-aligned documents. The basic architecture is a standard NMT system that uses the semantic information encoded in the internal representations of the network to determine at training time if an input sentence pair is *parallel enough* or not, and therefore whether it should be used for training or not. The selection is made online, so, the more the semantic representations improve during training, the more truly parallel sentence pairs are selected. Because of this, the nature of the selected pairs evolves during training, and this evolution is what we analyze as induced curriculum learning in Section 4. SS-NMT can be applied to different NMT architectures. Ruiter et al. (2019) showed good performance of SS-NMT both for RNN-based and transformer neural systems, differing in the internal semantic representations used to select the data. In this work, we focus on transformer architectures as they nowadays reach a higher translation quality (Barrault et al., 2019).

Let us assume a bidirectional NMT system $\{L1, L2\} \rightarrow \{L1, L2\}$ where the engine learns to translate simultaneously from a language $L1$ into another language $L2$ and vice-versa with a single encoder and a single decoder. This is important in the self-supervised architecture because it allows us to represent the two languages in the same semantic space. The input data to train the system is a monolingual corpus of sentences in $L1$ and a monolingual corpus of sentences in $L2$ and the system learns to select the adequate pairs. In order to speed-up the training, we use a comparable corpus such as Wikipedia, where we can safely assume that there are comparable (similar) and parallel sentence pairs in related documents $D_{L1}, D_{L2}$.

Given a document pair $D_{L1}, D_{L2}$, the SS-NMT system encodes each sentence of each document into two fixed-length vectors $C_w$ and $C_h$, such that

$$C_w = \sum_{t=1}^{T} w_t, \qquad C_h = \sum_{t=1}^{T} h_t, \tag{1}$$

where $w_t$ is the word embedding at time step $t$ and $h_t$ the encoder output. For each of the *sentence representations* referred to as $s$, all combinations of sentences $s_{L1} \times s_{L2} \| s_{L1} \in D_{L1}$ and $s_{L2} \in D_{L2}$ are encoded and then scored using the *margin-based* measure by Artetxe & Schwenk (2019b):

$$\text{margin}(s_{L1}, s_{L2}) = \frac{\text{sim}(s_{L1}, s_{L2})}{\text{avr}_{kNN}(s_{L1}, P_k)/2 + \text{avr}_{kNN}(s_{L2}, Q_k)/2}, \tag{2}$$

| | SS-NMT | | | | | | SOTA | |
| | L1–L2 | | | L2–L1 | | | L1–L2 | L2–L1 |
| | BLEU | TER | METEOR | BLEU | TER | METEOR | BLEU | BLEU |
| *en–fr* | 29.48 | 51.94 | 57.35 | 27.69 | 53.39 | 64.22 | 45.6/25.1/36.2 | –/24.2/33.5 |
| *en–de* | 14.40 | 69.28 | 39.95 | 18.84 | 62.15 | 55.13 | 35.0/17.2/22.5 | –/21.0/22.5 |
| *en–es* | 28.57 | 52.60 | 55.63 | 28.43 | 54.09 | 63.86 | –/–/– | –/–/– |

Table 2: BLEU, TER and METEOR scores of the self-supervised NMT systems trained on the three *en–{fr, de, es}* comparable WPs and tested on NT13/NT14. We compare the performance with current state-of-the-art (SOTA) systems in supervised NMT (Edunov et al.) / unsupervised NMT (Lample et al., 2018b) / unsupervised NMT+SMT (Artetxe et al., 2019).

where $\mathrm{avr}_{\mathrm{kNN}}(X, Y_k)$ corresponds to the average similarity between a sentence $X$ and $k\mathrm{NN}(X)$, its $k$ nearest neighbors $Y_k$ in the other language:

$$\mathrm{avr}_{\mathrm{kNN}}(X, Y_k) = \sum_{Y \in k\mathrm{NN}(X)} \frac{\mathrm{sim}(X, Y)}{k}. \tag{3}$$

What follows is a selection process, that identifies the top scoring $s_{L2}$ for each $s_{L1}$ and vice-versa. If a pair $\{s_{L1}, s_{L2}\}$ is top scoring for both language directions *and* for both sentence representations, it is accepted without the addition of any hyperparameter or threshold. This is the high precision, medium recall approach in Ruiter et al. (2019) and we use $k = 4$ as in Artetxe & Schwenk (2019b). Whenever enough pairs have been collected to create a batch, the system trains on it, updating its weights afterwards and proceeding to fill the next batch.

## 3.1 TRANSLATION QUALITY

**Experimental Setup** We use Wikipedia (WP) as a comparable corpus to train the self-supervised system. We download the English, French, German and Spanish WP dumps[1], pre-process them and extract the comparable articles per language pair using Wikitailor (Barrón-Cedeño et al., 2015). All articles are normalized, tokenized and truecased using standard Moses (Koehn et al., 2007) scripts. For each language pair, a shared byte-pair encoding (BPE) (Sennrich et al., 2016) of $100\,k$ merge operations is applied. The number of sentences, tokens and average article length is reported in Table 1. For validation we use *newstest2012* (NT12) and for testing *newstest2013* (NT13) for *en-es* and *newstest2014* (NT14) for *en–{fr, de}*. We use an OpenNMT-based (Klein et al., 2017) implementation of SS-NMT. Our models follow transformer base as defined in Vaswani et al. (2017). All systems are trained on a single GPU GTX TITAN using a batch size of 50 sentences with maximum sentence length being 50 tokens.

Monolingual embeddings trained using `word2vec` on the complete WP editions are projected into a common multilingual space via `vecmap` (Artetxe et al., 2017) to attain bilingual embeddings between *en-{fr,de,es}*. These embeddings initialise the word embeddings ($C_w$) in our system.

Finally, as a control experiment and purely in order to analyse the quality of the data selection auxiliary task, we use the Europarl (EP) corpus (Koehn, 2005). The corpus is pre-processed in the same way as WP, and we create a synthetic comparable corpus from it as explained in Section 3.2. For these experiments, we use the same corpora for validation and testing as mentioned above.

**Automatic Evaluation** Translation quality is evaluated by means of three automatic metrics: BLEU (Papineni et al., 2002), TER (Snover et al., 2006) and METEOR (Lavie & Agarwal, 2007). SS-NMT translation performance training on the *en-{fr, de, es}* comparable Wikipedias is reported in Table 2 together with a comparison to the current state-of-the-art (SOTA) in supervised and unsupervised NMT(+SMT). SS-NMT is on par with the current SOTA in unsupervised NMT, slightly outperforming it by ∼3-4 BLEU points in *en–fr* while having a lower performance on *en–de* (∼3 BLEU). Notice that unsupervised systems such as Lample et al. (2018b) use more than $400M$ monolingual sentences for training while we use an order of magnitude less by exploiting comparable corpora. However, once unsupervised NMT is combined with statistical methods, these outperform SS-NMT by large margins, i.e. ∼6 for *en–fr* and ∼5–9 BLEU for *en–de*. In a recent study

---

[1]Dumps were downloaded on January 2019 from `https://dumps.wikimedia.org/`

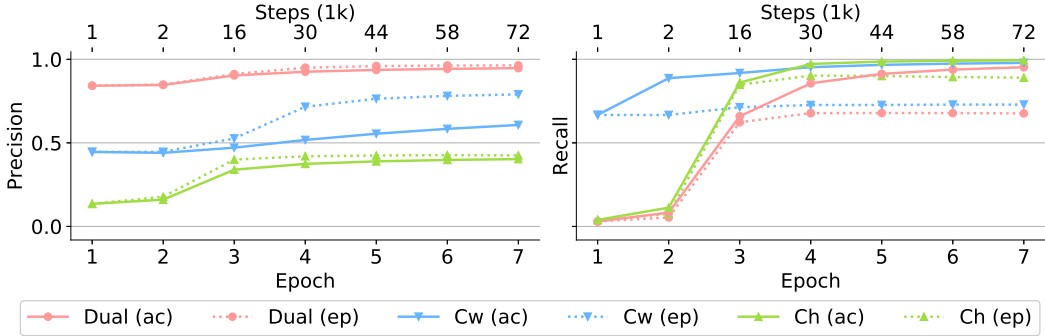

Figure 1: Accumulated (ac) and epoch-wise (ep) precision and recall on the $en-fr$ EP-based control experiments. A non-linearity on training steps per epoch can be observed in the $x$-axis.

by Wu et al. (2019), online data extraction from comparable documents has been combined with unsupervised methods for NMT, which allows for the additional exploitation of large amounts of monolingual data. They achieve similar BLEU scores as SS-NMT for $en2es$ (28.1) and $es2en$ (27.6) on NT13; notably SS-NMT trains on comparable data only.

## 3.2 DATA EXTRACTION QUALITY

**Experimental Setup**    To get an idea of the extraction performance of an SS-NMT system, we perform control experiments on synthetic comparable corpora, as there is no underlying ground truth to Wikipedia. For these purposes, we use the $en-\{fr,de,es\}$ versions of EP. After setting aside $1M$ parallel pairs as *true* samples to evaluate the extraction performance on, all remaining pairs are scrambled to create negative samples (*false*). In order to keep the synthetic comparable corpora close to the statistics of the original comparable WPs, we control the true:false sentence pair ratio and mimic what we observe in our extractions from WP. To estimate it, we start with the assumption that all accepted sentences are true (parallel) examples, and that the false examples (non-parallel) can be obtained as the total number of sentences minus the accepted ones. With this, we expect base true:false ratios of 1:4 for $en-\{fr,es\}$ and 1:8 for $en-de$. The negative samples are oversampled in order to meet this ratio given that there are $1M$ true samples. Further, we calculate the average article length of the comparable WPs and split the mixed corpus of true and false samples into articles. The detailed statistics of the synthetic pseudo-comparable EPs are reported in Table 1. We extract and train the SS-NMT system on these synthetic corpora after having initialized each model with the same bilingual embeddings as before.

**Automatic Evaluation**    The extracted pairs from EP at each epoch are compared to the $1M$ ground truth pairs to calculate the *epoch-wise* extraction precision (P) and recall (R). Further, we also take the concatenation of all extracted sentences up to a certain epoch in training in order to report the *accumulated* P and R. As we are interested in the final extraction decision based on the votes of both representations $C_w$ and $C_h$ (*dual*), but also in the decisions of each single representation ($C_w$, $C_h$), we report the performance for all three representation combinations on EP$_{enfr}$ in Figure 1. Similar curves can be observed for EP$_{ende}$ and EP$_{enes}$, which are considered in the discussion below.

At the beginning of training, the extraction **precision** of each representation itself is rather low with P$\in$[0.45,0.66] for $C_w$ and P$\in$[0.14,0.40] for $C_h$. The fact that $C_w$ is initialized using pre-trained embeddings, while $C_h$ is not, leads to this large difference in initial precision between the two. However, as both representations are combined via their intersections, the final decision of the model is high precision already at the beginning of training with values around $\sim 0.78-0.87$. As training progresses and the internal representations are adapted to the task, the precision of $C_h$ is greatly improved, leading to an overall high precision extraction which converges at $\sim 0.96-0.99$. This development of extracting parallel pairs with increasing precision can be viewed as a type of denoising curriculum as described by Wang et al. (2018).

The **recall** of the model, being bounded by the performance of the weakest representation, is very low at the beginning of training (R$\in$ [0.03,0.04]) due to the lack of task knowledge in $C_h$. However, as training progresses and $C_h$ improves, the accumulated extraction recall of the model rises to high values of $\sim 0.95-0.98$. Interestingly, the epoch-wise recall is much lower than the accumulated,

| | en–fr | | | en–de | | | en–es | | |
|---|---|---|---|---|---|---|---|---|---|
| | #Pairs | $en2fr$ | $fr2en$ | #Pairs | $en2de$ | $de2en$ | #Pairs | $en2es$ | $es2en$ |
| $\text{NMT}_{init}$ | 2.14M | 21.80 | 21.07 | 0.32M | 3.44 | 4.88 | 2.51M | 26.98 | 28.85 |
| $\text{NMT}_{mid}$ | 3.14M | 28.91 | 26.61 | 1.13M | 11.18 | 15.70 | 3.96M | 28.33 | 27.92 |
| $\text{NMT}_{end}$ | 3.17M | 28.80 | 26.48 | 1.18M | 11.89 | 15.91 | 3.99M | 28.34 | 28.09 |
| $\text{NMT}_{all}$ | 5.38M | 26.79 | 25.24 | 2.21M | 11.63 | 15.52 | 5.41M | 27.85 | 27.62 |
| SS-NMT | 5.38M | 29.48 | 27.69 | 2.21M | 14.40 | 18.84 | 5.41M | 28.57 | 28.43 |
| Wikimatrix | 2.76M | 33.50 | 30.12 | 1.57M | 13.61 | 12.17 | 3.38M | 29.60 | 28.30 |

Table 3: BLEU scores of a supervised NMT system trained on *Wikimatrix* as well as the unique pairs collected by SS-NMT in the first ($\text{NMT}_{init}$), intermediate ($\text{NMT}_{mid}$), final ($\text{NMT}_{end}$) and all ($\text{NMT}_{all}$) epochs of training. Evaluation performed on NT14 ($en$-$\{fr,de\}$) and NT13 ($en$-$es$) respectively.

which provides evidence for the hypothesis that SS-NMT models visit different *relevant* samples at different points in training, such that it has visited most of the samples at some point during training, but not at every epoch.

It should be stressed that the successful extraction of increasingly precise pairs in combination with high recall is the result of the dynamics of both internal representations $C_w$ and $C_h$. As $C_h$ is less informative at the beginning of training, $C_w$ guides the final decision to ensure high precision; and as $C_w$ is high in recall throughout the training, $C_h$ ensures a gentle growth in final recall by setting a good lower bound. The intersection of both ensures that errors committed by one have a chance of being caught by the other; a mutual supervision between representations.

**Comparison** We compare on the $en - \{fr, de, es\}$ corpora provided by Wikimatrix (Schwenk et al., 2019), which we pre-process as described in 3.1. As these consist of a collection of mined sentence pairs together with their similarity scores, a manual threshold $\theta$ needs to be set to extract pairs. We run the extraction script using $\theta = 1.04$, which has been suggested to be a *good choice for most language pairs*.[2] We then use the resulting corpus to train a supervised NMT system using the same specifications as in 3.1.

The results can be viewed in table 3. For $en$-$fr$, the supervised system trained on *Wikimatrix* outperforms SS-NMT by 3-4 BLEU points, while the opposite is the case for $en$-$de$, where SS-NMT achieves 1-5 BLEU points more. For $en$-$es$, both approaches perform similarly. The variable performance of the two approaches may be due to the varying appropriateness of the extraction threshold $\theta$ in *Wikimatrix*. This yields the problem that for each language and corpus, a new optimal threshold needs to be found; a problem that SS-NMT overcomes by its use of two representation types that compliment each other during extraction without the need of a manually set threshold.

## 4 SELF-INDUCED CURRICULA

### 4.1 ORDER & CLOSENESS TO THE TRANSLATION TASK

As a first indicator of the existence of a preferred choice in the order of the extracted sentence pairs, we compare the performance of SS-NMT with different supervised NMT models trained on the data extracted by SS-NMT at different points in training. We consider the first ($\text{NMT}_{init}$), intermediate ($\text{NMT}_{mid}$) and final ($\text{NMT}_{end}$) epochs of training as well as the concatenation of all unique pairs ($\text{NMT}_{all}$). We then train four supervised NMT models on this data. The difference in the **translation quality** one can obtain using only the data selected at different epochs illustrates the closeness of these data to the final translation task: we expect data extracted towards the end of the SS-NMT training to include more sentences which are parallel —as demanded by a translation task— and therefore to achieve a higher translation quality.

For each language pair and system, Table 3 shows the number of sentence pairs used for training and the achieved BLEU score. The SS-NMT training outperforms all the supervised versions across all tested languages. Notably, the performance is 1–3 BLEU points above the supervised system trained

---

[2]https://github.com/facebookresearch/LASER/tree/master/tasks/WikiMatrix

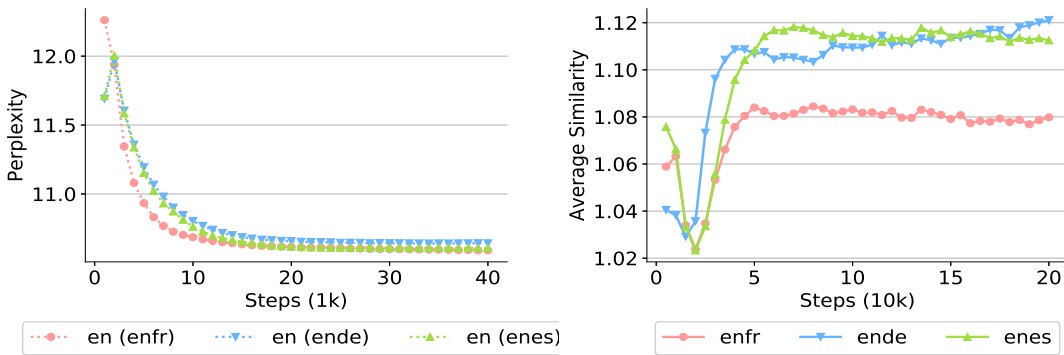

Figure 2: Perplexities on the English data extracted by SS-NMT (left) and average similarity scores of the accepted pairs (right).

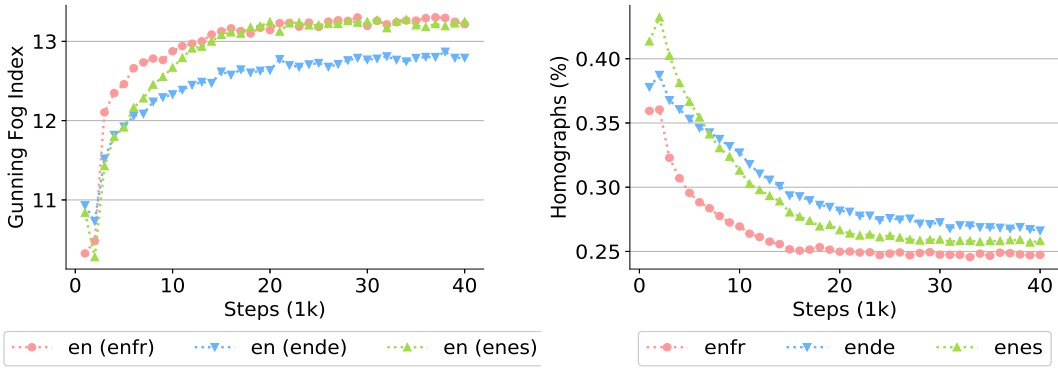

Figure 3: Gunning Fog Index (left) and percentage of homographs (right) of extracted English data seen during the first $40\,k$ steps in training.

on all extracted data, despite the fact that the SS-NMT system uses a much smaller amount of data in its first epochs. This suggests that the SS-NMT system is able to exclude previously accepted false positives at later epochs, while training directly on the entirety of its extracted data leads to a recurring visitation of the same erroneous samples. Further, the quality and quantity of the extracted data grows as training continues for all languages analysed. This can be observed by the fact that the concatenation of the data extracted across epochs (NMT$_{all}$) is always outperformed by the last and thus largest epoch (NMT$_{end}$), despite being much larger in size.

A second indicator of the closeness of the curriculum to the final task is of course the **similarity** between the selected sentence pairs during training. In our case, we estimate similarity between pairs by their margin-based scores (Eq. 2) during training. At the beginning of training, the average similarity between extracted pairs is comparatively low, but it quickly rises within the first $100\,k$ training steps to values close to $margin{\sim}1.07$ and $margin{\sim}1.12$ depending on the language pair. This evolution is depicted in Figure 2 (right). The increase in the mean similarity of the accepted pairs provides evidence for the base hypothesis that internal representations of translations grow closer in the cross-lingual space, and the system is able to exploit this fact by extracting increasingly similar —and accurate as seen in Section 3.2— pairs.

## 4.2 ORDER & COMPLEXITY

Establishing the complexity of a sentence is a complex task by itself. Complexity can be estimated by the loss of an instance with respect to the gold or target. In our self-supervised approach, there is no target for the sentence extraction task, so we try to infer complexity by other means.

First, we study the behaviour of the average **perplexity** throughout training. Perplexities of the extracted data are estimated using a language model trained with KenLM (Heafield, 2011) on the

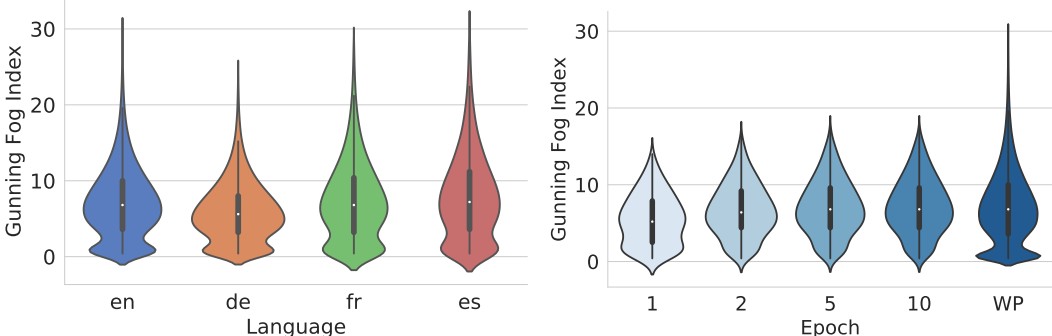

Figure 4: Kernel density estimated Gunning Fog distributions and box plots over the monolingual Wikipedias (left) and over extracted *en* (*en–de*) sentences at different points in training (right). For comparison, the distribution over monolingual $\text{WP}_{en}$ is included in the latter case.

monolingual WPs for the four languages in our study. We observe the same behaviour in the four cases which can be illustrated by the English curves plotted in Figure 2 (left). Perplexity drops heavily within the first $10\,k$ steps for all languages and models. This indicates that the data extracted in the first epoch includes more *outliers*, and the distribution of extracted sentences moves closer to the average observed in the monolingual WPs as training advances. The larger number of outliers at the beginning of training can be attributed to the larger number of homographs and short sentences (see Figure 3 (right)) at the beginning of training, leading to a skewed distribution of selected sentences.

The presence of **homographs** is vital for the self-supervised system in its initialization phase. At the beginning of training, only word embeddings, and therefore $C_w$, are initialized with pre-trained data, while $C_h$ is randomly initialized. Thus, words that have the same index in the vocabulary — homographs— play an important role in identifying similar sentences using $C_h$, making up around 1/3 of all tokens observed in the first epoch. As training progresses, and both $C_w$ and $C_h$ are adapted to the training data, the prevalence of homographs drops and the extraction is now less dependent on a shared vocabulary.

Finally, we analyze the complexity of the sentences that a SS-NMT system selects at different points of training by measuring their **readability**. For this, we apply a modified version of the **Gunning Fog Index** (GF) (Gunning, 1952), which is a measure of the years of schooling needed to understand a written text given the complexity of its sentences and vocabulary. It is defined as:

$$\text{GF} = 0.4 \left[ \left( \frac{w}{s} \right) + 100 \left( \frac{c}{w} \right) \right], \tag{4}$$

where $w$ and $s$ are the number of words and sentences in a text. $c$ is the number of *complex words*, which are defined as words containing more than 2 syllables. The original formula excluded several linguistic phenomena from the *complex word* definition such as compound words, inflectional suffixes or familiar jargon; we do not apply all the language-dependent linguistic analysis.

Since our training data is build up with Wikipedia articles, the diversity in the complexity of the sentences is limited to the range of complexities observed in Wikipedia. Figure 4 (left) shows the Gunning Fog distributions over the monolingual WPs. We plot the probability density function for the Gunning Fox Index for the four WP editions estimated via a kernel density estimation. We observe that each distribution is made up of two overlapping distributions; one with low sentence complexity containing article titles and headers, and one with a higher average complexity and larger standard deviation containing content sentences.

To study the behaviour during training, we compare the Gunning Fog distributions of the English data extracted at the beginning, middle and end of SS-NMT$_{ende}$ with that of the original mono $\text{WP}_{en}$. In the extracted data, we observe that the overlapping distributions are less pronounced and that there is no trail of highly complex sentences. This is due to (*i*) the pre-processing of the input data, which removes sentences containing less than 6 tokens, thus removing most WP titles and short sentences, and (*ii*) the length accepted in our batches, which is constrained to 50 tokens per sentence, removing highly complex strings. Apart from this, the distributions in the middle and the end of training come close to the underlying one, but we observe a large shift towards very

simple sentences in the first epoch. This shows that the system extracts mostly simple content at the beginning of training, but soon moves towards complex sentences that were previously not yet identifiable as parallel.

A more detailed evolution is depicted in Figure 3 (left). We accumulate extracted sentences within the period of $1\,k$ steps each and report their GF scores. Here we observe how the complexity of the sentences extracted rises strongly within the first $20\,k$ steps of training. For English, most models start with text that is deemed to be suitable for high school students (grade 10–11) and quickly turns to more complex sentences suited for undergraduate students in their first year ($\sim$13 years of schooling). The mean of the full set of sentences in the English Wikipedia is of $\sim$12, which corresponds to a high school senior. For all other languages, a similar trend of growing sentence complexity can be observed.

## 5 Summary and Conclusions

This paper explores self-supervised NMT systems where learning the MT model is done simultaneously with the selection of parallel sentences. This association makes the system define its own curriculum. We observe that the dynamics of mutual-supervision of both system's internal representations, $C_w$ and $C_h$, is imperative to the high recall and precision extraction of SS-NMT. Their combination for data selection over time resembles a denoising curriculum architecture in that the percentage of unprecise pairs —i.e. non-translations— decreases from 18% to 2%, with an especially fast descend at the beginning of training.

Even if the quality of the extraction increases with time, the lower-similarity sentences used at the beginning of the training are still relevant for the translation engine. We analyze the translation quality of a supervised NMT system trained on the extracted data at each epoch of the SS-NMT and observe a continuous increase in BLEU which cannot only be accounted for by the varying numbers of extracted pairs per epoch. Analogously, we also analyze the similarity scores of extracted sentences and see that they also increase over time. As extracted pairs are increasingly similar — and precise as argued before— the extraction itself resembles a secondary curriculum of growing task-relevance, where the task at hand is NMT learning with parallel sentences.

A tertiary curriculum of increased sample complexity could be observed via an analysis of the extracted data's Gunning Fog indexes. Here, the system starts with sentences suitable for high school students (GF=10–11) and quickly moves towards content suitable for first year undergraduate students (GF=13); an overachiever indeed as the norm over the complete WP is around GF=12.

Lastly, by estimating the perplexity with an external language model trained on WP, we observe a steep decrease at the beginning of training with fast convergence. This indicates that the extracted data quickly starts to resemble the underlying distribution of all WP data, with a larger amount of outliers at the beginning. These outliers can be accounted for by the importance of homographs at the beginning of training. This raises the question of how SS-NMT will perform on really distant languages, which is something that will need to be examined in our future work.

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
