# OpenReview forum: "Self-Induced Curriculum Learning in Neural Machine Translation"
_ICLR.cc/2020/Conference — Reject_

### Official Review · AnonReviewer2 · 2019-10-19
**Official Blind Review #2**

**Rating:** 3

**Review:**

** Paper summary **
Self-supervised machine translation (SS-NMT) is a problem where extracting data and training an NMT model are simultaneously conducted. (Ruiter et al., 2019) proposed several rules to select data and train models. This paper analyzes the following aspects of self-supervised machine translation (SS-NMT):
1.	Data extraction quality: precision and recall increase w.r.t. training iterations.
2.	Closeness to translation tasks: For the extracted sentences, as training goes on, the complexity decreases to the average level of potential bilingual corpus; the similarly of extracted sentences becomes closer. The authors also find that a joint process of extracting data and training models outperforms training a model with the extracted data.
3.	Complexity and similarity: The extracted sentences become harder w.r.t training epochs (measured by Gunning Fog Index). The presence of homographs becomes weaker and weaker.

** Details **
1.	The analysis is solid, but the findings are in general not quite surprising to readers. Besides, how should we leverage the findings in the paper?
2.	For the results in Table 3, what if we use all the data discovered by the initial, middle and end epochs (which might be duplicated) instead of ``unique’’ data?
3.     Can you show more statistics about data extraction time, training time?
4.     The relation with [ref1] should be discussed.

[ref1] Machine Translation with Weakly Paired Documents, https://openreview.net/pdf?id=ryza73R9tQ

**Experience Assessment:**

I have published in this field for several years.

**Review Assessment: Checking Correctness Of Derivations And Theory:**

I assessed the sensibility of the derivations and theory.

**Review Assessment: Checking Correctness Of Experiments:**

I assessed the sensibility of the experiments.

**Review Assessment: Thoroughness In Paper Reading:**

I read the paper thoroughly.

---

> ### Author Response · Authors · 2019-11-07
> **Findings, non-unique training, training time and related work**
>
> Dear Reviewer,
>
> thank you very much for your valuable comments. Let us address your concerns:
>
> 1.) To the best of our knowledge this study is the first deep analysis of self-supervised MT. It shows that a form of active selection naturally emerges from the setup, without having been explicitly programmed. This is a contribution to knowledge and increases our understanding of self-supervised learning, in the special setting presented here. We expect the detailed analyses and results to point to avenues for tackling data settings and situations where our current self-supervised approaches struggle, and therefore to help improving and extending the current approach. For example, the fact that homographs are important during the initialization phase of the model, SS-NMT systems may struggle with distant languages and differences in the writing systems. Future work could then build upon this.
>
> 2.) If we use all the data discovered throughout the SS-NMT training without duplicate removal to train a supervised NMT system, the supervised model would see exactly the same data in the same order as the SS-NMT system, leading to the same performance with slight statistically irrelevant variations. The point of extracting the unique data from the beginning, middle, end and all of the SS-NMTs training and then training a supervised system on this data, was to show that (i) the data quality increases from beginning to end and (ii) the order of the data matters (i.e. using SS-NMT as a simple data extractor for an external supervised system by training on all the data found by SS-NMT but not in exactly the same order and frequency is not optimal. Thus, the curriculum found by SS-NMT is beneficial for training).
>
> 3.) Data extraction and training happens simultaneously in the model. Training (with included data selection) of a large SS-NMT model on Wikipedia takes about 2~4 weeks on a single GPU GTX TITAN. Let us also report the number of epochs/steps/hours both SS-NMT and supervised baseline NMT (NMT_all) models were trained for:
>
> lang        SS-NMT                NMT_all
> en-fr:     7/420k/746h        7/750k/207h
> en-de:  10/210k/464h        10/440k/123h
> en-es:    6/450k/736h        6/650k/188h
>
> In both cases, the model is a Transformer base, i.e. a 6-layer encoder-decoder with
> 8-head self-attention, 512-dim word embeddings and a 2048-dim hidden
> feed-forward.
>
>
> 4.) Thank you for letting us know about this paper, which we would like to add to our "Related Work" section as it is indeed relevant to the topic.
>
> Best regards,
>
> The Authors

---

### Official Review · AnonReviewer3 · 2019-10-23
**Official Blind Review #3**

**Rating:** 1

**Review:**

This paper studies how to extract/select suitable training data from comparable —rather than parallel— corpora. The idea sounds reasonable.

My major concern is about the evaluation: it didn't compare with any existing work. Actually there quite a few papers  on mining parallel sentences from comparable corpora such as Wikipedia, as shown below. Seems the authors are not aware of those works and didn't review and compare with them. Without such comparisons, it is difficult to judge the effectiveness of the proposed method and the quality of this work.
[1] Finding similar sentences across multiple languages in Wikipedia, Proceedings of the Workshop on NEW TEXT Wikis and blogs and other dynamic text sources. 2006.
[2] Method for building sentence-aligned corpus from wikipedia, 2008 AAAI Workshop on Wikipedia and Artificial Intelligence (WikiAI08). 2008.
[3] Extracting parallel sentences from comparable corpora using document-level alignment, Human Language Technologies: The 2010 Annual Conference of the North American Chapter of the Association for Computational Linguistics. Association for Computational Linguistics, 2010.
[4] "Improving machine translation performance by exploiting non-parallel corpora." Computational Linguistics2006.
[5] https://www.aclweb.org/anthology/W04-3208.pdf
[6] https://openreview.net/pdf?id=ryza73R9tQ

Minor issues:
	1. "For each language pair, a shared byte-pair encoding (BPE) (Sennrich et al., 2016) of 100k merge operations is applied." Most papers on neural machine translation don't use such a large BPE size, which is likely to lead to better performance. It would be better to use the same setting as previous work for fair comparisons.

	2. "In the case of SS-NMT, both tasks —data extraction and learning NMT— enable and enhance each other, such that this mutual supervision leads to a self-induced curriculum, which is the subject to our analysis." Similar idea, mutual boosting between data selection and model training, has been explored in the following paper, although not for machine translation. What's the difference between these two papers?
Learning to Teach, ICLR 2018.

**Experience Assessment:**

I have published in this field for several years.

**Review Assessment: Checking Correctness Of Derivations And Theory:**

N/A

**Review Assessment: Checking Correctness Of Experiments:**

I carefully checked the experiments.

**Review Assessment: Thoroughness In Paper Reading:**

I read the paper at least twice and used my best judgement in assessing the paper.

---

> ### Author Response · Authors · 2019-11-07
> **Evaluation, BLEU and Related Work**
>
> Dear Reviewer,
>
> thank you very much for your valuable comments. Let us address first your main concern regarding the evaluation:
>
> We are aware of the wide range of prior research that has been done in the field of parallel data mining (e.g. on Wikipedia). However, the SS-NMT method we analyze in this paper does not intend to be a parallel data mining approach. Instead, it is a data selection method that depends on the models state, and thus may also select non-parallel sentences (i.e. similar pairs) if they are useful for the system. In this study, we analyze which kind of sentences are selected at different stages during training and we see, for example, that at the beginning of training non-parallel sentences can still be useful for learning.
> Nevertheless, we see the point that SS-NMT can be viewed as a data mining approach in itself. In order to capture this, we would add a section to the "Related Work" section to address this.
>
> As for a direct comparison of a data selection method on Wikipedia, we have recently performed experiments on the Wikimatrix corpus in en-{fr, de, es} [1], where a similar extraction method was used. We would be happy to add this experiment to this paper to compare the Wikimatrix approach to the SS-NMT approach. Let us report the BLEU scores we get from this experiment, where we trained a supervised NMT system on the corresponding Wikimatrix corpora:
>
> L1-L2        Wikimatrix        SS-NMT
> en-fr        33.50        29.48
> fr-en        30.12        27.69
> en-de        13.22        14.40
> de-en        12.17        18.06
> en-es        29.60        28.57
> es-en        26.63        26.42
>
> Here, Wikimatrx outperformed SS-NMT for en-fr, while SS-NMT is stronger in en-de, while the difference between the two methods is rather small for en-es.
>
> [1] Schwenk et al. 2019 "WikiMatrix: Mining 135M Parallel Sentences in 1620 Language Pairs from Wikipedia" https://arxiv.org/pdf/1907.05791.pdf
>
> Now to address your minor concerns:
>
> 1.) It is true that we use a rather large BPE size. However, it is the BPE value that was reported in the original SS-NMT paper, which is why we kept it for comparison. Nevertheless, you are right that BPE size is an interesting value for SS-NMT. High-resourced supervised NMT tends to performs better with larger BPE sizes, but SS-NMT also depends heavily on homographs during the beginning of training. Having a smaller BPE size can lead to more tokens being shared between two languages (taken that they are not distant languages, as is the case here for en-{fr, de, es}). If this decreased BPE size would then lead to a better initialization of SS-NMT and thus improved translation performance, would be something to investigate in future research.
>
> 2.) Thank you for bringing this paper to our attention, which we would like to add to the "Related Work" section. The main difference between the idea in "Learning to Teach" (LTT) and the SS-NMT approach is that LTT uses two separate models, a "teacher" and a "learner", which in a reinforcement setting mutually boost each other. However, in SS-NMT the "teacher" and the "learner" are the same model, and the data selection depends on the model state itself.
>
>
> Best regards,
>
> The Authors

---

### Official Review · AnonReviewer4 · 2019-11-08
**Official Blind Review #4**

**Rating:** 6

**Review:**

This paper describes a method for training self-supervised neural machine translation systems from a document-aligned comparable corpus (Wikipedia in en, fr, de and es).

The proposed training method consists of two concurrent processes: a pseudo-parallel sentence pair extraction process, where average word embeddings and encoder states are used to construct sentence embeddings which are compared to extract candidate sentence pairs, and a conventional model optimization process that uses online batches of the extracted sentence pairs as training data.

Experimental results on automatically evaluated translation quality on standard test sets are reported, in addition to parallel sentence extraction quality evaluated on the Europarl corpus and additional analyses on the self-induced curriculum resulting from the training process.

The proposed methodology is solid. The main issue with the paper is the lack of proper baseline comparison. The authors compare only with supervised and unsupervised systems trained on different corpora, and not with other approaches based on pseudo-parallel data extraction from Wikipedia.

EDIT:

I have increased my score based on the author's response.

**Experience Assessment:**

I have published in this field for several years.

**Review Assessment: Checking Correctness Of Derivations And Theory:**

N/A

**Review Assessment: Checking Correctness Of Experiments:**

I carefully checked the experiments.

**Review Assessment: Thoroughness In Paper Reading:**

I read the paper thoroughly.

---

> ### Author Response · Authors · 2019-11-13
> **Comparison**
>
> Dear Reviewer,
>
> thank you very much for your feedback and valuable comments. Let us address your concern regarding the comparison with other pseudo-parallel data extraction methods from Wikipedia:
>
> The SS-NMT method is a general data selection approach that may also select non-parallel sentences (i.e. similar pairs) if they are beneficial for its learning. For example, in our analysis of the selected sentences we saw that non-parallel sentences are selected in higher quantities at the beginning of training and allow the model to initialize itself on the task. However, we see how SS-NMT can also be seen as a (pseudo-) parallel data extraction method itself. To reflect this, we would like to add two parts to the revised version of our paper:
>
> i) a section describing previous work in the field of parallel data mining on Wikipedia will be added to "Related Work"
>
> ii) a comparison of SS-NMT with a supervised model trained on the en-{fr, de, es} Wikimatrix [1] corpora with optimal threshold. We have recently performed these experiments and would like to report the results here:
>
> L1-L2        Wikimatrix        SS-NMT
> en-fr        33.50        29.48
> fr-en        30.12        27.69
> en-de        13.22        14.40
> de-en        12.17        18.06
> en-es        29.60        28.57
> es-en        26.63        26.42
>
> The method used in [1] is quite similar to the base-idea of SS-NMT, with the main differences being that [1] only performs data mining and does this in a highly multilingual setting. This makes the comparison for us especially interesting.
>
> Nevertheless, we want to stress that the main focus of this study is the curriculum that arises in SS-NMT and how it is beneficial for learning. To combine this with the data mining perspective, it would be interesting to see how a supervised system would perform when training on the Wikimatrix data sampled according to a curriculum similar to that of SS-NMT, i.e. increasing complexity, decreasing noise etc. Unfortunately, because of the time constraints, we cannot report the results of this experiment during the rebuttal period, but they could be added to a potential camera ready version.
>
> [1] Schwenk et al. 2019 "WikiMatrix: Mining 135M Parallel Sentences in 1620 Language Pairs from Wikipedia" https://arxiv.org/pdf/1907.05791.pdf
>
> Best regards,
>
> The Authors

---

> > ### Comment · AnonReviewer4 · 2019-11-15
> > **Different data sources**
> >
> > WikiMatrix uses a different dump of Wikipedia and different extraction scripts than the one you used, for a completely fair comparison you should run the WikiMatrix extraction scripts with the LASER embeddings on your corpus.
> >
> > However, in my opinion the experiments that you provided, even with somewhat different corpora, show that you can obtain translation systems of comparable quality, and since you don't depend on any parallel resource your approach is potentially more widely applicable.

---

### Decision · Program_Chairs · 2019-12-19

**Decision:**

Reject

**Comment:**

This paper presents a method for curriculum learning based on extracting parallel sentences from comparable corpora (wikipedia), and continuously retraining the model based on these examples. Two reviewers pointed out that the initial version of the paper lacked references and baselines from methods of mining parallel sentences from comparable corpora such as Wikipedia. The authors have responded at length and included some of the requested baseline results. This changed one reviewer's score but has not tipped the balance strongly enough for considering this for publication.